# Generation of precision microstructures based on reconfigurable photoresponsive hydrogels for high-resolution polymer replication and microoptics

Pang Zhu [1], Qingchuan Song[1,2], Sagar Bhagwat[1], Fadoua Mayoussi[1], Andreas Goralczyk[1], Niloofar Nekoonam [1], Mario Sanjaya[3], Peilong Hou[1], Silvio Tisato [4], Frederik Kotz-Helmer[1,2,3], Dorothea Helmer [1,2,3,4] ✉ & Bastian E. Rapp[1,2,3,4]

Microstructured molds are essential for fabricating various components ranging from precision optics and microstructured surfaces to microfluidics. However, conventional fabrication technology such as photolithography requires expensive equipment and a large number of processing steps. Here, we report a facile method to fabricate micromolds based on a reusable photoresponsive hydrogel: Uniform micropatterns are engraved into the hydrogel surface using photo masks under UV irradiation within a few minutes. Patterns are replicated using polydimethylsiloxane with minimum feature size of 40 μm and smoothness of $R_q$ ~ 3.4 nm. After replication, the patterns can be fully erased by light thus allowing for reuse as a new mold without notable loss in performance. Utilizing greyscale lithography, patterns with different height levels can be produced within the same exposure step. We demonstrate the versatility of this method by fabricating diffractive optical elements devices and a microlens array and microfluidic device with 100 μm wide channels.

Micromolds play an important role in polymer replication to, e.g., fabricate surfaces with special wetting properties[1–3], microfluidic devices[4–7], high-sensitivity sensors[8,9], or mirooptics[10–12]. However, photolithography, which is the most widely used method to generate micromolds, suffers from several disadvantages, most prominently the multiple-step nature of the process, the requirement for sophisticated devices and the fact that larger amounts of toxic waste is produced during processing[13]. Therefore, versatile methods for the creation of inexpensive molds are highly sought-after. Hydrogels, owing to their easily tailorable physical and chemical properties, exhibit promising application potentials in various engineering fields, and have been used as molds for polymer replication. For example, Hirama et al.[14,15] and Sugiura et al.[16] manually arranged hydrogel molds and covered them using polydimethylsiloxane (PDMS). After curing PDMS, the hydrogel molds were washed out using hot water to fabricate microchannels. Similarly, by embedding flexible gel wires with high mechanical strength in PDMS solution and consecutively pulling out the gel templates from the cured PDMS matrix, three-dimensional (3D) microchannels with different cross sections were fabricated. However, in these methods, the gel templates need to be manually arranged, which is inconvenient and suffers from low precision. Using pre-gel solution as the photoresist, Dang et al.[17] prepared hydrogel molds for

[1]Laboratory of Process Engineering, NeptunLab, Department of Microsystems Engineering (IMTEK), Albert Ludwig University of Freiburg, Freiburg, Germany. [2]Freiburg Center of Interactive Materials and Bioinspired Technologies(FIT), Albert Ludwig University of Freiburg, Freiburg, Germany. [3]Glassomer GmbH, Freiburg, Germany. [4]Freiburg Materials Research Center (FMF), Albert Ludwig University of Freiburg, Freiburg, Germany. ✉e-mail: dorothea.helmer@neptunlab.org

polymer replication via mask-assisted photopolymerization, which are suitable for large area fabrication, but the over-exposure between adjacent structures resulted in limited resolution. Overall, owing to its flexibility which enables easy demolding and smooth surface, exploring better ways to utilize hydrogel as micromolds for polymer replication is a very promising approach.

Responsive hydrogels can be controlled by external stimulus and can exhibit biomimetic behaviors which has gained significant attention from the scientific community, e.g., for the creation of soft robots[18–21], wearable electronics[22,23], and medical devices[24,25]. Particularly, the use of soft materials like hydrogels, swollen polymer networks for generation of microstructures leads to Gaussian feature[26,27]. Such profiles are useful for generating smooth gradients, e.g. optical elements[28,29] as well as biomimetic structures for cells[30], which often require curved surfaces. Among them, photoresponsive hydrogels are particularly attractive because light as the stimulus exhibits many advantages. Light patterns can be generated at high resolutions and in very controllable fashion, are non-invasive, and offer the possibility for fast stimulus switches (on/off) with virtually no delay[31–33]. Volume change is a common responsive behavior of photoresponsive hydrogel under visible (VIS) or ultraviolet (UV) light. Overall, there are two pathways for preparing and utilizing photoresponsive hydrogels. Firstly, the macroscopic shape change of the hydrogel upon exposure can be used to generate actuated components for soft robotics[34–36], reversible microfluidic valves[37,38], and controllable drug delivery system[39]. Secondly, mask-assisted selective irradiation can be used to generate hydrogels with switchable surface topographies. As an example, Kuenstler et al.[40] fabricated a photoresponsive hydrogel based on the host-guest complex between cyclodextrins and azobenzene groups which shows reversible wetting property changes owing to isomerization[41–43]. Via photo-induced localized swelling/deswelling of the network, the hydrogel sheet changed its topography reversibly. Stumpel et al.[44] firstly prepared a hydrogel containing spiropyran with varying crosslinking density. After swelling, a patterned hydrogel surface was obtained due to the different crosslinking-density-dependent swelling abilities. Under visible light exposure, the topography of the surface changed as a result of photoinduced deswelling of the hydrogel network. In addition, Han et al. recently demonstrated that laser projection via a digital micromirror device (DMD) can directly change the network structure of polyacrylate-based hydrogels, which is subsequently utilized to guide the assemble of nanoparticles to form designed 2D or 3D structures showing promising application as optical microdevices, like diffractive optical elements (DOE)[45]. We reasoned that similar approaches should be applicable for generating micromolds similar to the conventional processing pipeline using photolithography. In photolithography, a photosensitive polymer (i.e., a photoresist) is structured using selective irradiation, mostly using photomasks. Depending on the nature of the photoresist (positive- or negative-type), the irradiated areas (in positive-type photoresists) or the non-exposed areas (in negative-type photoresists) can be removed in a subsequent development step. The obtained physical structure is then used as a micromold for the generation of, e.g., microoptical components such as lenses or lens arrays. The disadvantage of this workflow is that the structure, once defined, cannot be changed and a different micromold must be generated for every design. In contrast to this, we reasoned that a photoswitchable hydrogel surface, where a surface topography could be generated by photo-induced localized swelling/deswelling as a consequence of light exposure should allow the generation of a reconfigurable micromold, i.e., a mold which can be set and reset multiple times (Fig. 1a). We refer to these reusable and reconfigurable micromolds as micromold displays reflecting their ability to be dynamically set and reset.

In this work, we present a facile method to generate micromold displays based on the photoresponsive acrylamide/azobenzene-cyclodextrin (AM/AZO-CD) hydrogel prepared via thermal radical polymerization and their usage in polymer replication. Micromolds with different profiles and sizes can be formed easily and quickly on the hydrogel surface and replicated with high fidelity onto PDMS components. In contrast to classical photolithography, this process requires no development step, drying or subsequent etching processes or substances such as potassium hydroxide or hydrofluoric acid which are standard in classical micromold generation. Owing to the reversibility of azobenzene-cyclodextrin (AZO-CD) host-guest interaction[36,40], the patterned hydrogel can be reused at least four times by erasing the original structure using visible light. Utilizing a custom-made lithography system based on a DMD, patterns with a wide variety of feature sizes and geometries can be realized directly on the hydrogel surface at low cost and within a few minutes. Using a grayscale digital light processing (DLP), grayscale lithography can be conveniently carried out to prepare micropatterns with different heights in one step. Furthermore, we demonstrate the applicability of the method by fabrication of a microfluidic device and a microlens array and high-performance diffractive optical elements (DOE) using the micromold display

## Results

### Micromold display fabrication

The photoresponsive AM/AZO-CD hydrogel is fabricated via thermal radical polymerization. Firstly, the organic gel is synthesized via co-polymerization of acrylamide (AM) and 4-methacryloyloxyazobenzene (AZO). 2,2'-azobis(2-methylpropionitrile) (AIBN) and N,N'-methylene-bisacrylamide (MBA) are chosen as the thermal initiator and cross-linker, respectively. The obtained gel (Supplementary Fig. 2a) is then immersed in α-cyclodextrin (α-CD) solution for solvent exchange. During this process, trans-azobenzene forms a complex with α-CD because of host-guest interaction[40,46]. Due to the hydrophilic outer surface of the α-CD unit, the network becomes more hydrophilic once the α-CD units cover the hydrophobic azobenzene units. In this state, the gel absorbs more water and reaches a fully-swollen state (Fig. 1b) with a swelling ratio of about 55 wt% (Supplementary Fig. 2b). Tran-cis isomerization of azobenzene happens upon illumination with UV light. Because of its different spatial structure, cis-azobenzene groups slip out from α-CD cavity, leaving more hydrophobic constituents exposed to the surrounding water environment. As a result, water is partially repelled from the network resulting in a shrunken hydrogel (Fig. 1b). A free thin hydrogel film shrinks by $16.9 \pm 0.6$ vol% in-plane (X/Y) and $16.3 \pm 0.7$ vol% out-of-plane (Z-direction, see Supplementary Fig. 3a & b). The trans-cis isomerization was confirmed via ultraviolet-visible (UV-VIS) absorption spectroscopy of the AM/AZO-CD hydrogel (Supplementary Fig. 4a & b). The intensity of the characteristic absorption peak of trans-azobenzene groups at 323 nm decreases gradually when illuminating the hydrogel with UV light (320–400 nm, $4.6 \ mW \ cm^{-2}$) and recovers to original intensity under VIS light illumination (400-700 nm, $4.9 \ mW \ cm^{-2}$). After five cycles, the absorption peak at 323 nm shows little change (below 6%) exhibiting good reversibility and stability (Supplementary Fig. 4c).

To enable high-resolution micromold displays, the gel is prepared as a uniform, smooth film of 1 mm thickness on a glass slide. For doing so, two glass slides are spaced with an adjustable spacer (made from silicone rubber) and the AM/AZO-CD hydrogel is prepared in between them (Supplementary Fig. 5). The bottom glass slide is functionalized with 3-methacryloxypropyl dimethylchlorosilane (MACS) to covalently attach the polymerized gel onto the glass substrate. The polymerized gel is stored in α-CD solution for 24 h to reach swelling balance. Subsequently, the gel is exposed to UV light (320–400 nm or 300–400 nm, Fig. 1a), structured either by a physical mask or by an in-house developed maskless projection lithography system based on a DMD device[47]. The illuminated areas of the hydrogel become more hydrophobic, which causes local shrinkage in the hydrogel (Fig. 1a, b) and results in a microstructured gel surface and thus a micromold. The

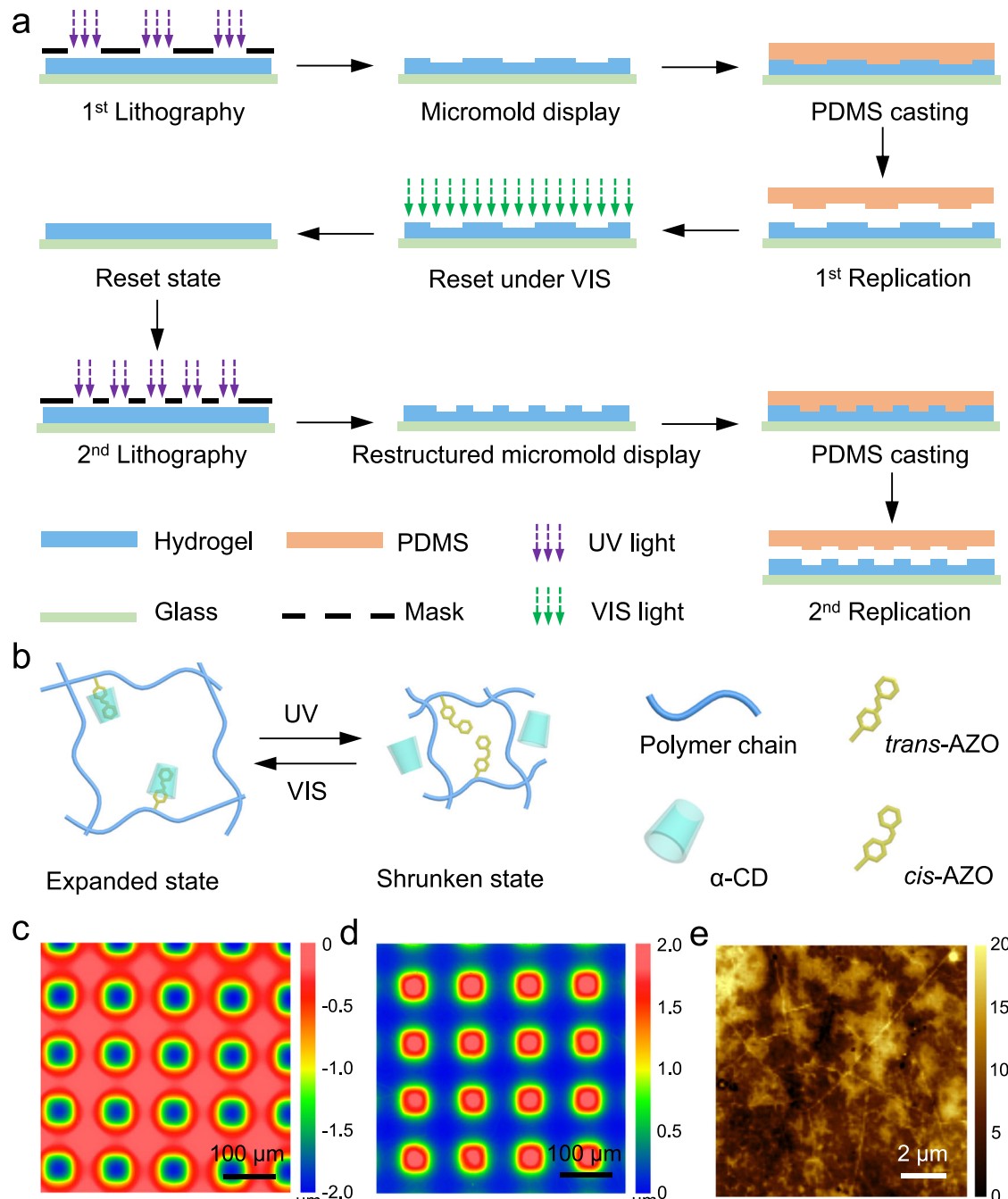

**Fig. 1 | Preparation of AM/AZO-CD hydrogel based micromold displays and their subsequent usage in a replication process. a** Structuring and replication process on the acrylamide-azobenzene/cyclodextrin (AM-AZO/CD) hydrogel based micromold display; (**b**) Mechanism of light responsive AM-AZO/CD hydrogel; (**c**) Surface profile of the micromold, i.e., the structured hydrogel after 1 min of structured UV irradiation, (**d**) PDMS component replicated from the micromold; (**e**) the PDMS surface characterization shows high smoothness with a $R_q$ ~ 3.4 nm.

microstructure on the gel was analyzed by white light interferometry (WLI). After 1 min mask-assisted UV irradiation, a uniform microstructure is generated on the hydrogel surface (Fig. 1c). To emphasize the stability of the engraved structure on the gel, it was left in α-CD solution without extra protection to prevent ambient light and analyzed again 1 h after UV illumination. No detectable changes were observed in the gel structure (Supplementary Fig. 6). In order to demonstrate the ability of this structure to function as a micromold, PDMS replication was carried out. For this, water on the hydrogel surface is gently removed using a nitrogen gun. The PDMS precursor is then poured onto the microstructured hydrogel surface. After curing,

the microstructured PDMS can be easily peeled off from the hydrogel micromold. The replicated PDMS structure was also analyzed by WLI (Fig. 1d). Cross-sectional profiles of the hydrogel micromold and the replicated PDMS were compared (Supplementary Fig. 7), and the data shows that the microstructure on the hydrogel surface is replicated with high fidelity with features as fine as 40 μm. The surface generated shows low surface roughness with a $R_a$ of 2.68 nm and $R_q$ of 3.4 nm (Fig. 1e) as determined by atomic force microscopy (AFM). In addition, the prepared hydrogel can be stored for long time in the glass cell (at least 5 months) without noticeable loss the performance, which is desirable in practical applications. To verify this, a micro-square array

was structured onto the freshly prepared hydrogel and the hydrogel stored for 5 months using a physical mask (illumination time 10 s), the obtained structures show no difference (Supporting information Fig. 8).

**Repeated setting and resetting of the micromold display**

As stated, the reversible nature of the gels allows micromold displays to be structured, reset and structured using different patterns. In order to allow multiple setting/resetting cycles, the chemical composition of the gels was investigated and adjusted. A balance between gel stiffness for easy handling and gel softness for increased depth of the generated microstructures, as well as illumination time for fast processing, was sought. By adjusting the crosslinker (MBA) molar ratio between 1.2 mol%, 1.6 mol%, and 2 mol% in the gel, the elastic modulus of the

hydrogel was modulated from 18 kPa to 27 kPa, as determined by AFM (Fig. 2a). A low modulus as in a soft gel of 1.2 mol% crosslinker enables the formation of deeper microstructures, with a depth of 10 μm formed within 5 min of illumination for a square array pattern (width 100 μm, gap 100 μm) (Fig. 2b). However, higher softness causes the hydrogel vulnerable to adhesion force between the hydrogel and the cover slide when opening the glass cell (Supplementary Fig. 5) in which the gel was polymerized, which relatively makes it more difficult to obtain hydrogel with high surface quality. In contrast, higher crosslinking density (up to 2 mol% of crosslinker ratio) allows gels with higher elastic modulus and improved mechanical resilience, but reduced engraving speed (Fig. 2b). As a compromise, a gel with 1.6 mol% crosslinker ratio was chosen. For this variant, an engraving depth of 10.3 μm after 15 min exposure could be achieved (Fig. 2c)

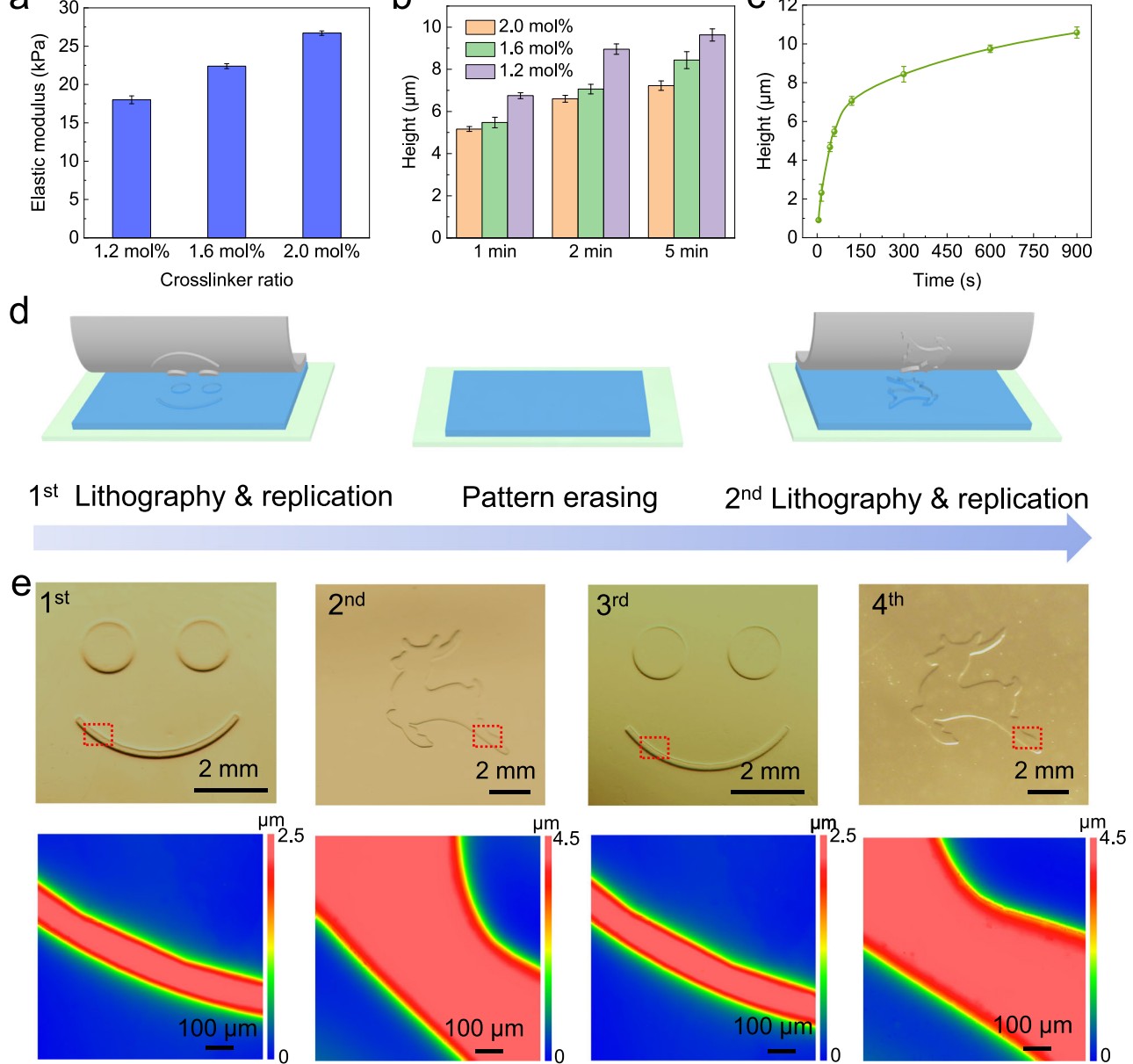

**Fig. 2 | Optimization and usage of the micromold display. a** Increasing crosslinker ratio results in AM/AZO-CD hydrogels with a higher elastic module; (**b**) Higher crosslinker ratio causes a decrease of the engraving speed on the hydrogel, but facilitate handing of the micromold displays; consequently, the hydrogel formulation with 1.6 mol% crosslinker was chosen for further experiments; (**c**) Engraving height increases with increasing exposure time, reaching 10.6 μm after 15 min UV irradiation; (**d**) Illustration of consecutive setting/resetting cycles of the micromold display with different topographies; (**e**) Optical pictures and WLI characterization of four PDMS substrates replicated from micromolds generated on the micromold display with a smiley (1st and 3rd) and a deer (2nd and 4th) structure. Data in a, b and c are presented as mean values ± SD. Error bars represent the standard deviation from three samples.

while providing excellent mechanical resilience during manufacturing and better handling of the micromold displays.

The reversibility of the micromold display was demonstrated by consecutive PDMS replications (Fig. 2d). A first pattern was engraved on the micromold display and consecutively used for PDMS replication. The display was then immersed into α-CD solution and illuminated for 1 h under VIS light (400–700 nm). After 48 h of storing in the dark, the generated pattern was completely erased and the display was thus reset. Subsequently, a second pattern was generated, replicated and erased in the same manner. The process was performed for 4 times, and pictures of consecutively replicated PDMS pattern and profiles based on WLI characterization of each replicated structures are shown in Fig. 2e, demonstrating the full reversibility of the setting as well as the reproducibility across several set/reset cycles.

In addition, the versatility of the hydrogel micro-molds was also demonstrated by engraving a butterfly structure on the curved cylindrical hydrogel surface, then, the engraved structure was successfully replicated using PDMS (Supplementary Fig. 9a, b), which is desirable in many engineering fields, like optical device fabrication[48].

## Micromold display via grayscale projection lithography

Maskless projection lithography enables high-resolution illumination without generation of a greyscale physical mask. Using a DMD, greyscale lithography is implemented using pulse-width modulation of the mirrors. Figure 3a illustrates the principle of using maskless projection lithography for the structuring of the micromold display using a custom-made DMD[47]. A triangle array pattern was created on the micromold displays, and replicated into a PDMS film (Fig. 3b). Benchmarking the lateral resolution on the DMD setup, we found that square arrays with a width and gap of 40 μm can be replicated with good fidelity. Smaller sizes caused structures to blend together and become indistinguishable (Supplementary Fig. 10). A square array pattern (width 100 μm, gap 100 μm) was used to test the engraving speed on the micromold display based on the DMD setup, the engraving depth of the gel quickly and linearly increased to 8 μm within roughly 90 s before slowly reaching to 11 μm after 10 min illumination (Fig. 3c). This phenomenon is, on the one hand, owing to the limited UV penetration ability, the very top hydrogel surface receives the highest exposure energy to induce the isomerization of azobenzene groups and results in rapid growth speed of the engraved structure. With increasing depth into the hydrogel, the received exposure energy reduced reduces gradually, and the growth speed decreased correspondingly. On the other hand, the contraction of the exposed area is constrained by the ambient polymer network, the constrain force increases with larger contraction, which also contribute the slower growing speed with increasing exposure time. Similar phenomenon has been reported when fabricating microstructures on a soft photo responsive substrate[26,27]. In addition to uniform structure arrays, digital masks with arbitrary structures, a tree for example, can be easily designed and used to engrave patterns on the micromold display, and then be replicated using PDMS (illumination time: 120 s, Fig. 3d). As demonstrated with the physical mask-based setup, the micromold displays could be repeatedly set and reset using the DMD setup as well (illumination time: 120 s, Fig. 3e, f). In addition, as shown in supplementary Fig. 11, the hydrogel shows no significant fatigue after 6 times structuring-erasing cycles, which shows its stable ability to be restructured as micromolds. The maskless DMD setup was further used for grayscale lithography which gives the ability to modulate the projected light intensity in spatially controlled fashion. This allows varying the local volumetric change on the micromold display as a function of the set grayscale value during the UV exposure. As an example, a square array was projected with different greyscale values (Supplementary Fig. 12) for the squares (Fig. 3g), resulting in patterns with different height for every single square (illumination time: 120 s). The SEM images of obtained

microstructures show decent surface quality (Supplementary Fig. 13). Utilizing an annular grayscale digital mask, a staircase structure of 6.7 μm with three height levels was fabricated within 20 s (Supplementary Fig. 14). Besides grayscale masks with uniform shapes and simple grayscale gradient, a multiple grayscale flower image (Fig. 3h) was used to generate a micromold with complex structure (illumination time: 30 s). The PDMS structure replicated from the micromold display shows the analogous flower structure with different heights following the grayscale image.

## Optical and microfluidic devices fabrication

Without requiring any sophisticated devices or hazardous chemicals, the developed micromold displays represent an easily accessible method for microstructuring and soft lithography, affording structuring such as, e.g., microoptical or microfluidic devices. As an example, a microlens array was generated by illuminating the micromold display via a circle array physical mask. The replicated PDMS microlens array shows uniform structures (Fig. 4a, b) and the intended clear diffraction pattern (laser source: 532 nm) (Fig. 4c). To further highlight the versatile application of the micromold display in optical engineering field, DOE devices with complex microstructured surface were fabricated to reconstruct arbitrary user-defined patterns. Firstly, a binary hologram (Fig. 4d) was generated using Gerchberg-Saxton (GS) algorithm[45,49,50] by inputting the designed image like a dolphin (inset of Fig. 4d) as the source file. The generated hologram was then used as the digital mask and input into the homemade projection lithography system for engraving structures onto the hydrogel surface for PDMS replication. Figure 4e shows the 3D profile of the replicated PDMS structure with feature resolutions of about 30 μm and the corresponding diffraction pattern (Fig. 4f, laser source: 650 nm). Holograms of various patterns such as a butterfly image and the IMTEK logo could be generated for engraving structures onto the hydrogel surface, and the replicated PMDS DOE devices showed high performance to display the designed patterns vividly (Supplementary Fig. 15).

In addition to the optical devices, we fabricated a digital mask for a microfluidic diffusive mixer structure for usage of the micromold display. The illuminated micromold display was then used as the master for PDMS replication. As shown in Fig. 4g and Fig. 4h, the obtained serpentine microchannel in the replicated PDMS exhibited uniform width (100 μm) and depth (20 μm) at different areas throughout the pattern. To test the performance of the channel, the PDMS replica was bonded onto a glass slide and dyed water with different colors was injected into the two mixer inlets. The diffusive mixing could be clearly observed throughout the meander channel structure (Fig. 4i). As an alternative, a microfluidic chip can be generated only requiring a single replication step which shows same performance although with lower depth (Supplementary Fig. 16).

## Discussion

In this paper, we developed a micromold display based on AM/AZO-CD photoresponsive hydrogels which allow the generation of masters, e.g., for polymer replication circumventing the laborious, cost- and waste-intensive workflows commonly associated with photolithography. The micromold display makes use of the localized contraction induced by spatially controlled UV illumination of the gel, which engraves microstructures into the hydrogel within a few minutes, thus significantly simplifying classic replication process workflows. Besides the generation of binary (single-height) microstructures with tens of micron resolution, the developed micromold display also allows for greyscale patterning, creating different height levels in one illumination step, which allows the generation of replication master structures with different height levels as required, e.g., for the fabrication of microlens arrays. We demonstrated the ability to design microlens arrays using grayscale lithography to generate gradient

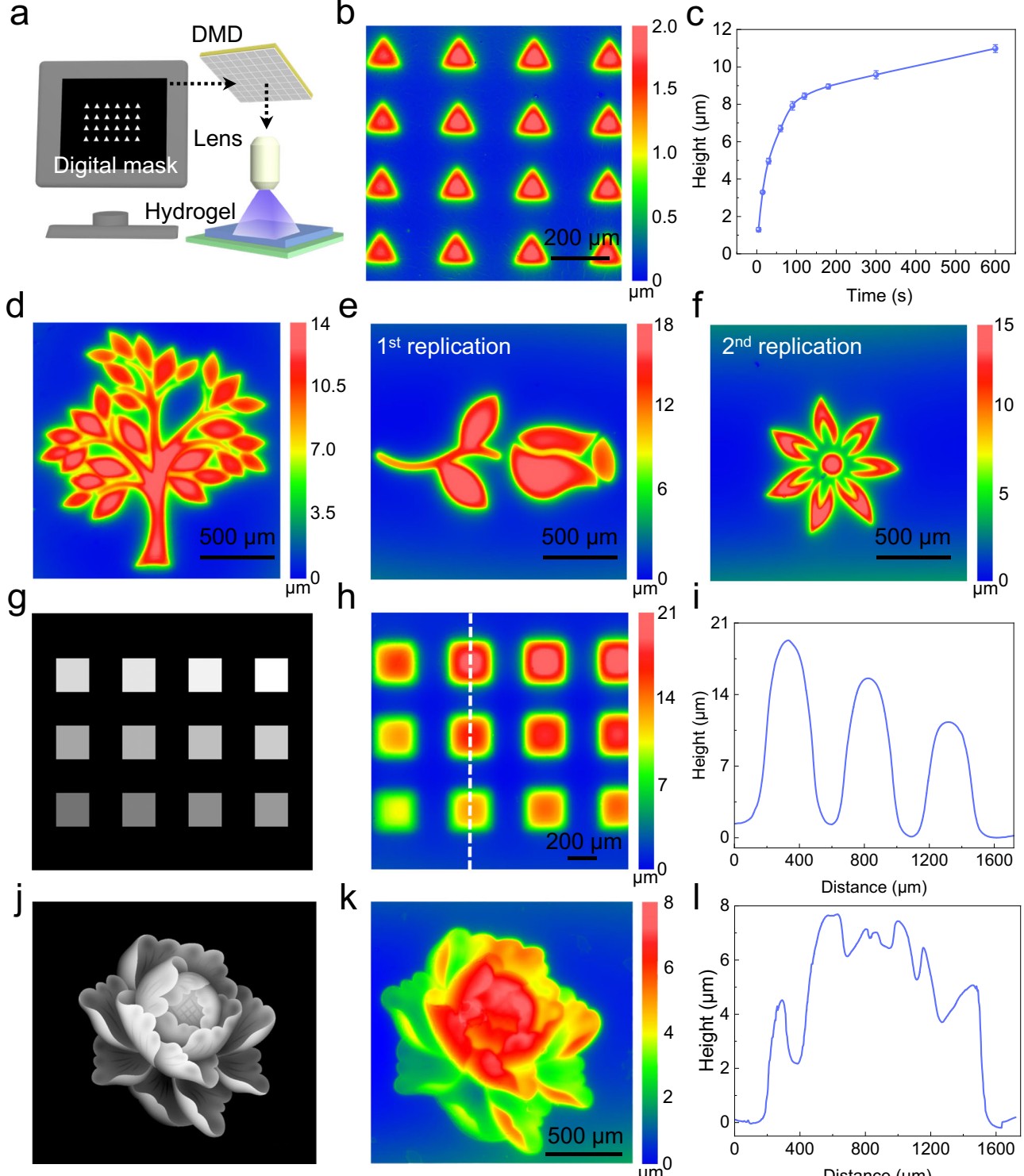

**Fig. 3 | Micromold display structures generated using digital light processing (DLP). a** Scheme of maskless projection lithography system based on a digital micromirror device (DMD); (**b**) WLI profile of replicated PDMS with a triangle array; (**c**) Engraving height increases over exposure time reaching 11 µm after 10 min of illumination; (**d**) The profile of replicated PDMS with a tree structure showing the ability of the micromold display to fabricate complex shapes; (**e, f**) Two consecutive lithography-replication process based on reversibility of the AM/AZO- CD hydrogel via digital masks using the custom-made DMD; (**g**) A square array greyscale digital mask used for grayscale lithography, and (**h, i**) WLI image and feature profile of obtained structure on the replicated PDMS with different height; (**j**) A grayscale flower mask used for grayscale lithography and (**k, l**) WLI image and feature profile of replicated flower on PDMS with different height based on the varied transparency of the mask. Data in c are presented as mean values ± SD. Error bars represent the standard deviation from three samples.

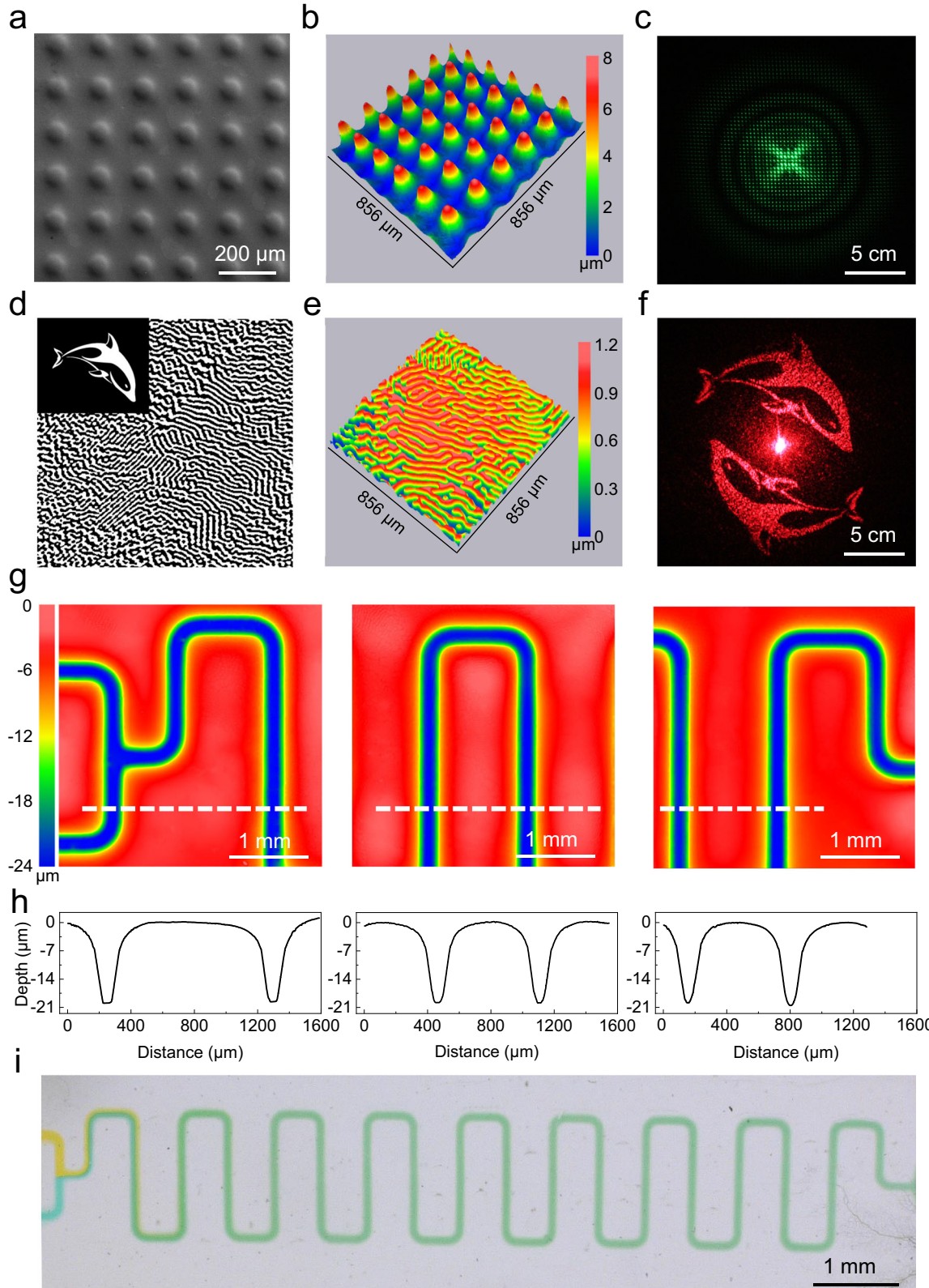

**Fig. 4 | Optical and microfluidic devices fabrication using the developed micromold display. a** Scanning electron micrograph and (**b**) 3D topography of PDMS microlens array and (**c**) corresponding diffraction pattern; (**d**) The binary hologram used as the digital mask for lithography using the Gerchberg-Saxton (GS) algorithm (the inset is original user-designed image that was used to generate the corresponding digital masks); (**e**) 3D profiles of the replicated PDMS DOE devices determined by WLI which show a feature resolution of about 30 µm and (**f**) corresponding diffraction patterns; (**g**) WLI characterization and (**h**) feature profile of the replicated microchannel in PDMS at different locations along the channel; (**i**) Optical picture of the microchannel filled with dyed water showing diffusive mixing.

patterns in the micromold displays; and high-performance DOE devices with complex microstructure are fabricated to display arbitrary user-designed patterns under coherent laser source. Furthermore, for demonstrating the capabilities of the micromold displays or the generation of high-resolution structures, a microfluidic device was fabricated with uniform width and depth throughout the whole channel. Usage of the micromold displays alleviates the necessity for expensive cleanroom-based photolithography, toxic solutions and cost-intensive photoresins. We believe that this platform can become a valuable tool for microengineering, biotechnology, clinical analytics and miniaturization, effectively providing access to a simple, reusable device for the generation of replication masters in polymer molding, soft lithography, UV casting and similar processes.

## Methods
### Materials
Acrylamide (AM, 99%), α-cyclodextrin (α-CD, 99%), 4-phenylazopheonl (98%), methacrylic anhydride (94%), 2,2'-azobis(2-methylpropionitrile) (AIBN, 98%), $N,N'$-methylenebisacrylamide (MBA, 99%), 4-dimethylaminopyridine (DMAP, 98%), sodium chloride (NaCl, 99%), magnesium sulfate ($MgSO_4$, 98%) were purchased from Sigma-Aldrich, Germany. Methanol (99.9%), dimethyl sulfoxide (DMSO, 99.5%), tetrahydrofuran (THF, 99.5%) were purchased from Carl Roth, Germany; 3-methacryloxypropyl dimethylchlorosilane (MACS, 92%) and $1H,1H,2H,2H$-perfluoroctyldimethylchlorosilane (97%) were purchased from Abcr GmbH, Germany. Polydimethylsiloxane (PDMS, 601 A and 601B) was purchased from Wacker, Germany. Transparent silicon rubber sheet was purchased from Technikplaza, Germany.

### Synthesis of 4-methacryloyloxy azobenzene monomer
4-methacryloyloxy azobenzene (AZO) monomer is synthesized according to a modified method as reported in literature[2]. 4-phenylazopheonl (7.92 g, 0.04 mol, 1 eq.), methacrylic anhydride (7.392 g, 0.048 mol, 1.2 eq.), DMAP (100 mg) were dissolved in 50 mL dried THF in a round-bottom flask. The mixture was stirred at 50 °C for 20 h and then cooled to room temperature. After being washed with saturated NaCl solution for three times, the oil phase was dried using $MgSO_4$ and evaporated to obtained the crude product. The crude product was recrystallized in methanol twice to obtained the pure AZO monomer (yield: 83%), and NMR result (250 MHz, Bruker, USA) shows no impurity (Supplementary Fig. 1).

### Preparation of AM/AZO-CD hydrogel
AM/AZO-CD hydrogels with different crosslinker content were prepared. As an example, AM/AZO-CD hydrogel with 1.6 mol% crosslinker was prepared dissolving 2130 mg AM, 240 mg AZO monomer, 73.92 mg MBA and 39.36 mg AIBN in DMSO to obtain a clear pre-gel solution. The solution was injected into a sandwich glass cell composed of a common glass slide, a MACS modified glass slide prepared based on a reported method[51] and a PDMS spacer (Supplementary Fig. 5). The polymerization was carried out in an oven (47° C) for 72 h. After removing the glass slide, the gel was immersed into α-CD/$H_2O$ solution (25 mg mL$^{-1}$) for 24 h to reach the fully-swollen state.

For 3D lithography on curved hydrogel surface, the pre-gel solution prepared above was injected into a glass tube (inner diameter: 15 mm). The glass tube was sealed and put into the oven (47 °C) for 72 h for polymerization. A cylindrical hydrogel was obtained by breaking the glass tube after polymerization, and was immersed into α-CD/$H_2O$ solution (25 mg mL$^{-1}$) for 24 h to reach the fully-swollen state.

The swelling ratio of the AM/AZO-CD gel was calculated according to the function: $SR = (Wt-Wo)/Wo$ X 100%. Where, SR is the swelling ratio, Wo is the original weight of the gel and Wt is the real-time weight measured after immersing the gel into α-CD for different times.

### Lithography of the micromold display, structure replication, and erasing of the displayed structure
For mask-based lithography, the hydrogel was immersed in α-CD/$H_2O$ solution and covered with the physical mask with a light intensity of 4.6 mW cm$^{-2}$ (wavelength: 320-400 nm). For maskless projection lithography, the digital pattern was projected onto the hydrogel via a custom-made maskless lithography system based on a DMD[47] using a light intensity of 3.2 mW cm$^{-2}$ at 300–400 nm. After exposure, remaining water on the AM/AZO-CD hydrogel surface was gently removed using a nitrogen gun. PDMS was mixed and degassed according the manufacturer and poured onto the micromold display surface and left to cure for 6 h at room temperature. Subsequently, the cured PDMS was peeled off from the hydrogel surface to obtain the replicated structure. After replication, the hydrogel was immersed in α-CD/$H_2O$ solution and illuminated with VIS light (400–700 nm, 4.9 mW cm$^{-2}$) for 1 h, and then stored in a dark environment for 48 h to recover its original state.

### Microfluidic device and microlens array fabrication
A microchannel pattern was first created on the micromold display using a physical mask under UV light (time: 5 min, 320–400 nm, 4.6 mW cm$^{-2}$) before replicating the displayed structure into PDMS to obtain the positive mold. The positive structured PDMS template was then treated using a handheld corona discharge (BD-20V, 230 V, 50/60 Hz, Electro-Technic Products Inc., USA)[52] for 30 s and immersed into the 1$H$,1$H$,2$H$,2$H$-perfluoroctyldimethylchlorosilane solution (3 v/v % in toluene) for 10 min. After being washed with isopropanol and completely dried in an oven at 100 °C for 30 min, another layer of PDMS resin was poured on the obtained positive PDMS model and peeled off after being cured. The obtained PDMS microchannel was washed with isopropanol and processed using corona discharge (30 s). It was then placed on a clean glass slide freshly treated using corona discharge (30 s) and cured under a pressure in an oven (100 °C) for 1 h.

The microlens structure was firstly engraved on the micromold display using a physical circle array mask (diameter/gap: 50/100 µm) under UV light (320-400 nm, 4.6 mW cm$^{-2}$) for 15 min. The structure was then replicated into PDMS using the described procedure.

### Characterization
UV-VIS absorption of AM/AZO-CD hydrogel was carried out using a UV-VIS spectrophotometer (Evolution 201, Thermofisher, Germany). The surface structure of the micromold display and the replicated PDMS was characterized using a white light interferometry (WLI, Newview 9000, Zygo, Germany) and scanning electron microscope (SEM, Tescan Amber X, Czech Republic). Youngs modulus of hydrogels was characterized using atomic force microscopy (AFM) (JPK Nanowizard 4, Bruker, Germany), and roughness of the replicated PDMS structures was characterized using AFM (Multimode 8, Bruker, Germany).

## Data availability
Source Data file has been deposited in Figshare under accession code DOI link: https://doi.org/10.6084/m9.figshare.25726524.

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

## Acknowledgements

This work is supported by the Deutsche Forschungsgemeinschaft (DFG, German Research Foundation) under Germany's Excellence Strategy – EXC–2193/1 – 390951807, the European Research Council (ERC) under the European Union's Horizon 2020 research and innovation program (grant agreement no. 816006), and the Research Cluster "Interactive and Programmable Materials (IPROM)" funded by the Carl Zeiss Foundation.

## Author contributions

B.E.R. and D.H. conceived the idea. P.Z. designed the experiments. P.Z., Q.S., S.B., F.M., A.G., N.N., M.S., P.H., and S.T. performed the experiments. P.Z. analyzed the results. P.Z., D.H., F.K., and B.E.R. drafted the manuscript, and all authors contributed to writing the manuscript.

## Funding

## Competing interests

The authors declare no competing interests.
