## [Peer Review File · Nature Communications]

Generation of precision microstructures based on reconfigurable photoresponsive hydrogels for high-resolution polymer replication and microopticsREVIEWER COMMENTS

Reviewer #1 (Remarks to the Author):

Zhu et al. present a method to produce microstructured molds for silicone (PDMS) molding in a photoresponsive hydrogel by patterned light illumination. The manufacturing of the photoresponsive hydrogel material and its ability to reversibly de-swell on local UV illumination and re-swell on white light illumination has previously been reported by Kuenstler et al. (ref. 35) using a DMD-based maskless lithography exposure system as also employed by Zhu et al. The novelty in the current manuscript is the application of the structured hydrogels to shape PDMS object surfaces, with the resulting PDMS objects directly functioning as devices or being used for further molding of second generation PDMS replicas.

The authors' approach to PDMS molding is viable, and their characterization method of using white-light interferometric microscopy is appropriate for the compliant hydrogel molds and PDMS replicas produced. However, the reported spatial resolution is much inferior to other photolithographic techniques commonly used for PDMS molds, such as patterned UV exposure of spin-coated SU-8. The latter can provide nearly vertical side walls for wall heights of hundreds of micrometers. In contrast, Zhu et al. show very gently sloping side walls with depth gradually changing over dozens of micrometers for a step height of less than two micrometers (Supplementary Fig. 7). This is equally apparent in the authors' approach to producing a microfluidic channel (Fig. 4g-i), where the nominally rectangular channel shape more closely resembles a gaussian channel profile.

Zhu et al. explore the use of timed UV exposure (Fig. 2) and their DMD-based maskless lithography system (Fig. 3) to produce grayscale lithography of controllable depth. This is an interesting concept that has also previously been investigated in SU-8 based photolithography for generation of topographically complex molds. The authors successfully demonstrate a correlation between exposure dose on the photoresponsive hydrogel and the height of protrusions in the replicated PDMS objects in Fig. 3. Specifically, a quantitative correlation is provided between exposure time and replica protrusion height in Fig. 3c. It is surprising that the correlation apparently is not smooth throughout across the investigated exposure times, specifically having a kink at 200 s. The authors do not discuss this surprising observation of the lack of smoothness. Obviously, such variation could be caused by measurement uncertainties, but the error bars on all measurements are exceedingly small. Importantly, they do not specify what the error bars represent or from how many independent experiments data were acquired to produce the error bars. This applies both to Fig. 2c and Fig. 3c, but especially Fig. 2c where the uncertainty on some data points appears to be on the order of 0.1 micrometer. The authors should include the raw data for these calibration data in a supplementary file for the reader to fully appreciate the reported high precision of the method.

The authors should also inform the reader on how the grayscale calibration data in Fig. 3g was analyzed, and what the experimental uncertainty is on the outcome. First, the height profile in Fig. 3g right-hand sub-figure clearly has a sloping background level with the valley between the two larger peaks not reaching the background level. Second, the exposure levels in Fig. 3g left-hand sub-figure monotonically increase in going left-to-right bottom-to-top, while the resulting replicated features decrease in height from the right-most square in a row to the left-most square in the row above. This is likely caused by inhomogeneous light exposure by the maskless illumination setup, also suggested by the skewed cross-sectional profile of the concentric exposure of increasing intensity shown in Supplementary Fig. 10. The authors should include an analysis of the illumination homogeneity and discuss how that can influence the reported accuracy of the generated recesses in the hydrogel.

The method presented is interesting as an alternative method for micromold manufacturing. However, its severely limited spatial resolution and limited range of depth makes it unlikely to replace existing methods for prototyping of molds, in particular patterned UV light exposure of thick resin materials such as SU-8. Traditionally, shadow masks would be needed to be procured from external suppliers for patterned resin exposure, but direct-write systems are increasingly commercially available. The authors say that it is a disadvantage that conventionally patterned

resins cannot be reused. However, the cost of the resin itself is usually not the cost-limiting factor in using photolithography for mold manufacturing, but rather the experimental setup to generate the patterned light. The authors also state that patterning on curved cylindrical surfaces "... is beyond the ability of conventional photolithography technology, which only works on a flat surface." This is correct if using contact lithography, but photosensitive resins on curved surfaces can obviously also be patterned by projection lithography systems similar to the setup used by the authors.

Reviewer #2 (Remarks to the Author):

In the present paper, the authors use a photo-responsive hydrogel as micro mold for PDMS molding. When irradiated with a UV light pattern the hydrogel shrinks locally, which allows micro structuring its surface. After being dried, the structured surface can be used for PDMS molding at room temperature. The microstructures can be later erased by being exposed to visible light, allowing the hydrogel to be reused. Simple optics and micro-fluidic devices were fabricated to demonstrate the versatility of the method.

This is a very nice and very original contribution to the existing literature, which stands out from the recent publications in this domain, and I am happy to support its publication.

Questions:

- 1- Line 163: Instead of using the word "modulus", it would help the readers to use "elastic modulus" here.
- 2- The energetic density used in the UV-irradiation and VIS-irradiation steps is indicated in the methods paragraph, but it would make sense to also indicate it in the main text when the different irradiation steps are discussed.
- 3- Line 237: "Figure 4g and Figure 4g"

Reviewer #3 (Remarks to the Author):

The paper reports on the synergic combination of responsive hydrogels with photomask-assisted UV irradiation or greyscale lithography for the low-cost fabrication of reconfigurable polymeric micromolds to create large area PDMS micropatterns by standard soft-lithography potentially exploitable for different fields of application mainly including microfluidics and microoptics. The real possibility to introduce novel smart fabrication process capable of decreasing time and costs typically required by standard lithography while maintaining the same resolution has a strong soundness in the micro/nanofabrication fields as also demonstrated by the big research efforts focusing on the smart combination of 3D-printing with novel materials.

Undoubtedly, the approach reported is very appealing, it uses low-cost materials and enables for micromolds easily replicable with PDMS. However, before considering the publication, I have some major comments mainly focused on some statements by the authors and with some doubts on the experimental parts. In particular, all the advantages they mentioned for the fabrication process (fastness, low-costs, high resolution), that are the core of the research, are not so evident in my opinion.

1) Fastness: The main point I would like to highlight is that the authors stated that the reported fabrication process is time-saving, especially considering that reconfigurable molds can be used to create many replicas. According to what reported in the Manuscript, summing up the times reported in the materials and methods section, at least six days are necessary to fabricate the polymer molds, engraving it, replicating with PDMS and reconfiguring the molds. Furthermore, these 6 days could be acceptable if the molds could be replicated many times, while the authors stated that they can be used for 4 replicas before degrading. Also from Figure 2 E, from the 2nd to the 4th replicas the shape differences as well as the different surface roughness are evident.

It is also important to take into account that the main proof-of-concepts demonstrating the potential applications of the reconfigurable molds for microfluidics and microlens arrays i) request the use of a physical mask to guarantee the resolution and ii) are obtained by two-steps PDMS replicas, where the 1st PDMS replica from the reconfigurable molds thus requiring an additional step of silanization and further extending fabrication times.

In addition, I have some doubts on the possibility to crosslink the PDMS at room temperature in only 6 hours. Standard procedure typically requires at least 24 hours when working at ambient temperature thus further increasing time for replica preparation.

2) Low cost procedure: If I well understood, to demonstrate the potential application of the reconfigurable molds for high-resolution microfluidic and microoptics, the engraving process is based on the UV-irradiation through physical masks. Are these masks fabricated by standard processes? If this is the case, then the presented approach still relies on side expensive masks fabrication processes when higher resolution is needed.

3) High-resolution: the authors stated that they can exploit the procedure for high-resolution polymer replication at lower costs. However, for proof-of-concept applications where the highest resolution is obtained, mask-assisted UV irradiation is needed, and I was wondering how these masks are produced. On the other hand, the resolution decreases when using DMD and digital masks especially with respect to reported literature and considering fabrication resolution that can be obtained by replica molding of polymeric stamps realized by effective low-cost rapid prototyping methods such as 3D printing. Also I am not so convinced on the surface roughness of the replicas as well on the exact shape of the arrays obtained by using the grayscale projection. I would suggest adding SEM images at least to each example of applications as reported for the microlens arrays.

I would like the authors to comment on these three aspects that I consider critical.

Regarding the experimental part, I would like to raise some question:

4) What is the difference in wettability between the cis and trans form? I think should be something already known in literature.

5) Based on experience of PDMS replicas of plastic molds, I am aware that when replicating from hydrogels or materials not completely cured, there could be a sticky effect that affects PDMS polymerization or that could require longer PDMS curing (again in contrast with the 6h curing process mentioned in 1)). Probably the authors observed this phenomenon as stated in lines 167-168. Furthermore, I would like to ask if the percentage of crosslinker could affect the speed of reconfiguration.

6) Minor comments:

- could you comment on the trend obtained in Figure 3c?
- the caption 4a and 4b should be inverted, and at line 223 the reference for figure 4f should be changed with 4c.

Reviewer 1

Zhu et al. present a method to produce microstructured molds for silicone (PDMS) molding in a photoresponsive hydrogel by patterned light illumination. The manufacturing of the photoresponsive hydrogel material and its ability to reversibly de-swell on local UV illumination and re-swell on white light illumination has previously been reported by Kuenstler et al. (ref. 35) using a DMD-based maskless lithography exposure system as also employed by Zhu et al. The novelty in the current manuscript is the application of the structured hydrogels to shape PDMS object surfaces, with the resulting PDMS objects directly functioning as devices or being used for further molding of second generation PDMS replicas.

Answer: We thank the reviewer for the constructive review of our work. All comments are addressed on a point-by-point basis below. We would like to point out that while Kuenstler *et al.* use the same effect of host/guest interaction, they merely show macroscopic bending of structures, and the reported method is not capable of high-resolution and grayscale structuring of hydrogel surfaces. Furthermore, the reported material system is different. While Kuenstler *et al.* use a NIPAM-based material which is sensitive predominantly via thermal (or photothermal) activation, we developed a material which is sensitive to light exposure.

Question 1. *The authors' approach to PDMS molding is viable, and their characterization method of using white-light interferometric microscopy is appropriate for the compliant hydrogel molds and PDMS replicas produced. However, the reported spatial resolution is much inferior to other photolithographic techniques commonly used for PDMS molds, such as patterned UV exposure of spin-coated SU-8. The latter can provide nearly vertical side walls for wall heights of hundreds of micrometers. In contrast, Zhu*

et al. show very gently sloping side walls with depth gradually changing over dozens of micrometers for a step height of less than two micrometers (Supplementary Fig. 7). This is equally apparent in the authors' approach to producing a microfluidic channel (Fig. 4g-i), where the nominally rectangular channel shape more closely resembles a gaussian channel profile.

Answer 1: While we agree that conventional lithography technology with SU-8 provides high resolution and its ability to create steep walls, it is a cleanroom process which requires a multiple step protocol processing with specialized machinery and, in the end, provides a single-geometry replication mold. A new design iteration would require a new lithography process to be carried out. The re-write ability we demonstrate in this paper is not possible in classical photolithography. Furthermore, there are many occasions where steep walls are not first choice and where gradients have to be approximated by multiple layers of many lithographic layers with small height steps. This multilayer lithography leads to a very long and multi-step process. Examples include the shape of microfluidic channels for biomimetic studies as well as the creation of optical devices such as lenses and diffractive optical elements. To make these advantages clearer to the reader, we have adjusted the introduction to emphasize the advantages of Gaussian profiles for microfabrication and we have included the fabrication of a diffractive optical element with complex topography and lateral resolution of about 30 μm (Fig. 4d and Supplementary Fig. 16) to show the practical use of the method for complex fabrication.

We have made the following changes to the manuscript:

We have included the following sentences to the introduction:

The use of soft materials like hydrogels, swollen polymer networks for generation of microstructures leads to Gaussian feature profiles^{26,27}. Such profiles are particularly useful for generating smooth gradients, e.g. optical

elements^{28,29} as well as biomimetic structures for cells³⁰, which often require curved surfaces.

We have included the following papers to the reference:

26 Chen, D. et al. Homeostatic growth of dynamic covalent polymer network toward ultrafast direct soft lithography. *7*, eabi7360, doi:doi:10.1126/sciadv.abi7360 (2021).

27 Li, T. et al. Hierarchical 3D Patterns with Dynamic Wrinkles Produced by a Photocontrolled Diels–Alder Reaction on the Surface. *32*, 1906712, doi:10.1002/adma.201906712 (2020).

28 Aldalali, B., Kanhere, A., Fernandes, J., Huang, C.-C. & Jiang, H. Fabrication of Polydimethylsiloxane Microlenses Utilizing Hydrogel Shrinkage and a Single Molding Step. *Micromachines* *5*, 275-288, doi:10.3390/mi5020275 (2014).

29 Dong, L., Agarwal, A. K., Beebe, D. J. & Jiang, H. Adaptive liquid microlenses activated by stimuli-responsive hydrogels. *Nature* *442*, 551-554, doi:10.1038/nature05024 (2006).

30 Esch, M. B., Post, D. J., Shuler, M. L. & Stokol, T. Characterization of in vitro endothelial linings grown within microfluidic channels. *Tissue engineering. Part A* *17*, 2965-2971, doi:10.1089/ten.tea.2010.0371 (2011).

We have included the following data to the results:

Fig. 4 Optical and microfluidic devices fabrication using the developed micromold display. (a) 3D topography and (b) scanning electron micrograph (a) Scanning electron micrograph and (b) 3D topography of PDMS microlens array and (c) corresponding diffraction pattern; (d) The binary hologram used as the digital mask for lithography using the Gerchberg-Saxton (GS) algorithm (the inset is original user-designed image that was used to generate the corresponding digital masks); (e) 3D profiles of the replicated PDMS DOE devices determined by WLI which show a feature

resolution of about 30 μm and (f) corresponding diffraction patterns; (g) WLI characterization and (h) feature profile of the replicated microchannel in PDMS at different locations along the channel; (i) Optical picture of the microchannel filled with dyed water showing diffusive mixing.

Question 2. *Zhu et al. explore the use of timed UV exposure (Fig. 2) and their DMD-based maskless lithography system (Fig. 3) to produce grayscale lithography of controllable depth. This is an interesting concept that has also previously been investigated in SU-8 based photolithography for generation of topographically complex molds. The authors successfully demonstrate a correlation between exposure dose on the photoresponsive hydrogel and the height of protrusions in the replicated PDMS objects in Fig. 3. Specifically, a quantitative correlation is provided between exposure time and replica protrusion height in Fig. 3c. It is surprising that the correlation apparently is not smooth throughout across the investigated exposure times, specifically having a kink at 200 s. The authors do not discuss this surprising observation of the lack of smoothness. Obviously, such variation could be caused by measurement uncertainties, but the error bars on all measurements are exceedingly small. Importantly, they do not specify what the error bars represent or from how many independent experiments data were acquired to produce the error bars. This applies both to Fig. 2c and Fig. 3c, but especially Fig. 2c where the uncertainty on some data points appears to be on the order of 0.1 micrometer. The authors should include the raw data for these calibration data in a supplementary file for the reader to fully appreciate the reported high precision of the method.*

Answer 2: We thank the reviewer for this valid observation. We have repeated all experiments to ensure consistency in the data. We have tested the feature depth upon different exposure time on three individual samples and three spots were test on each single sample. We find that the data is highly reproducible

and a kink was not observed. We have integrated the data and adjusted Figure 2c and 3c.

We have made the following changes to the manuscript:

We have updated Fig. 2b and Fig. 2c in results:

Fig. 2 Optimization and usage of the micromold display. (a) Increasing crosslinker ratio results in AM/AZO-CD hydrogels with a higher elastic modulus; (b) Higher crosslinker ratio causes a decrease of the engraving speed on the hydrogel, but facilitate handing of the micromold displays; consequently, the hydrogel formulation with 1.6 mol% crosslinker was chosen for further experiments, (c) Engraving height increases with increasing exposure time, reaching 10.3-10.6 μm after 15 min UV irradiation; (d) Illustration of consecutive setting/resetting cycles of the micromold display with different topographies; (e)

Optical pictures and WLI characterization of four PDMS substrates replicated from micromolds generated on the micromold display with a smiley (1st and 3rd) and a deer (2nd and 4th) structure.

We have updated Fig. 3c in the results and added sequence numbers to every single sub-figure:

Fig. 3 Micromold display structures generated using digital light processing

(DLP). (a) Scheme of maskless projection lithography system based on a digital micromirror device (DMD); (b) WLI profile of replicated PDMS with a triangle array;

(c) Engraving height increases over exposure time reaching 11 μm after 10 min of illumination; (d) The profile of replicated PDMS with a tree structure showing the ability of the micromold display to fabricate complex shapes; (e & f) Two consecutive lithography-replication process based on reversibility of the AM/AZO-CD hydrogel via digital masks using the custom-made DMD; (g) A square array grayscale digital mask used for grayscale lithography, and (h & i) WLI image and feature profile of obtained structure on the replicated PDMS with different height; (h) (j) A grayscale flower mask used for grayscale lithography and (k & l) WLI image and feature profile of replicated flower on PDMS with different height based on the varied transparency of the mask.

Question 3. *The authors should also inform the reader on how the grayscale calibration data in Fig. 3g was analyzed, and what the experimental uncertainty is on the outcome. First, the height profile in Fig. 3g right-hand sub-figure clearly has a sloping background level with the valley between the two larger peaks not reaching the background level. Second, the exposure levels in Fig. 3g left-hand sub-figure monotonically increase in going left-to-right bottom-to-top, while the resulting replicated features decrease in height from the right-most square in a row to the left-most square in the row above. This is likely caused by inhomogeneous light exposure by the maskless illumination setup, also suggested by the skewed cross-sectional profile of the concentric exposure of increasing intensity shown in Supplementary Fig. 10. The authors should include an analysis of the illumination homogeneity and discuss how that can influence the reported accuracy of the generated recesses in the hydrogel.*

Answer 3:

Firstly, Concerning the height distribution of figure 3g, we would like to clarify that every square has a different intensity and thus the resulting height of the structures is expected to be different. To make this more obvious to the reader,

we have explained this in the text more clearly and included an image of the greyscale values of the mask in the supporting information.

We have made the following changes to the manuscript:

We have included information of grayscale mask of Fig. 4g to the draft:

As an example, a square array was projected with different greyscale values (Supplementary Fig. 12) for the squares (Fig. 3g), resulting in patterns with different height for every single square.

We have included the following figure to supplementary information:

Supplementary Fig.12 Transparency analysis of grayscale mask employed in DMD grayscale lithography.

Secondly, the point raised to the sloping background is correct but this is not a feature of the illumination or the system but due to alignment during projection and WLI.

Question 4. *The method presented is interesting as an alternative method for micromold manufacturing. However, its severely limited spatial resolution and limited range of depth makes it unlikely to replace existing methods for prototyping of molds, in particular patterned UV light exposure of thick resin materials such as SU-8. Traditionally, shadow masks would be needed to be*

procured from external suppliers for patterned resin exposure, but direct-write systems are increasingly commercially available. The authors say that it is a disadvantage that conventionally patterned resins cannot be reused. However, the cost of the resin itself is usually not the cost-limiting factor in using photolithography for mold manufacturing, but rather the experimental setup to generate the patterned light. The authors also state that patterning on curved cylindrical surfaces "... is beyond the ability of conventional photolithography technology, which only works on a flat surface." This is correct if using contact lithography, but photosensitive resins on curved surfaces can obviously also be patterned by projection lithography systems similar to the setup used by the authors.

Answer 4: (1) The reviewer is certainly right that the resolution of classical SU-8 Technology is high. However, as the reviewer points out, the machinery to conduct the classical processes is expensive and is a definite limitation of the process. Here, our process offers an easily accessible method to fabricate micromolds. The advantage of material reuse is not only in saving money, but also should be considered under sustainability aspects, which are becoming increasingly important. Furthermore, as we pointed out also in Answer 1, we see the strong point of our method in the generation of reusable grayscale patterns, and we underline this by demonstrating the use of our material system for the generation of complex DOE (see also Answer1)

Taken together, we think our methods offers true value for flexible generation of micropatterns for various applications and offers an advance in the field.

(2) we agree that the photosensitive resins on the curved surface can be also patterned by projection lithography systems and that the statement might be confusing to the readers. We have rephrased the sentence to make it clearer.

We have made the following changes to the manuscript:

We have made the following change to the results:

In addition, the versatility of the hydrogel micro-molds was also demonstrated by engraving a butterfly structure on the curved cylindrical hydrogel surface, then, the engraved structure was successfully replicated using PDMS (Supplementary Fig. 8a & b 9a & b), This is beyond the ability of conventional photolithography technology, which only works on a flat surface, which is desirable in many engineering fields, like optical device fabrication⁴⁸.

We have included the following paper to the reference:

48 Zhang, D., Yu, W., Wang, T., Lu, Z. & Sun, Q. Fabrication of diffractive optical elements on 3-D curved surfaces by capillary force lithography. Opt. Express 18, 15009-15016, doi:10.1364/OE.18.015009 (2010).

Reviewer 2

In the present paper, the authors use a photo-responsive hydrogel as micro mold for PDMS molding. When irradiated with a UV light pattern the hydrogel shrinks locally, which allows micro structuring its surface. After being dried, the structured surface can be used for PDMS molding at room temperature. The microstructures can be later erased by being exposed to visible light, allowing the hydrogel to be reused. Simple optics and micro-fluidic devices were fabricated to demonstrate the versatility of the method.

This is a very nice and very original contribution to the existing literature, which stands out from the recent publications in this domain, and I am happy to support its publication.

Answer: We thank the reviewer for this positive assessment of our work.

Question 5. *Line 163: Instead of using the word "modulus", it would help the readers to use "elastic modulus" here.*

Answer 5: We have included the suggested correction in the text and the title of Y axis of the corresponding Fig. 2a (see Answer 2)

We have included the following change to the results:

By adjusting the crosslinker (MBA) molar ratio between 1.2 mol%, 1.6 mol% and 2 mol% in the gel, the **elastic** modulus of the hydrogel was modulated from 18 kPa to 27 kPa, as determined by AFM (Fig. 2a).

Question 6. *The energetic density used in the UV-irradiation and VIS-irradiation steps is indicated in the methods paragraph, but it would make sense to also indicate it in the main text when the different irradiation steps are discussed.*

Answer 6: The energetic density has been included into the main text for different irradiation steps.

We have included the following changes to the results:

The intensity of the characteristic absorption peak of *trans*-azobenzene groups at 323 nm decreases gradually when illuminating the hydrogel with UV light (320-400 nm, 4.6 mW cm⁻²) and recovers to original intensity under VIS light illumination (400-700 nm, 4.9 mW cm⁻²)

Question 7. Line 237: "*Figure 4g and Figure 4g*"

Answer 7: The mistake has been corrected.

We have included the following changes to the results:

As shown in Fig. 4g and Fig. 4gh, the obtained serpentine microchannel in the replicated PDMS exhibited uniform width (100 μm) and depth (20 μm) at different areas throughout the pattern.

Reviewer 3

The paper reports on the synergic combination of responsive hydrogels with photomask-assisted UV irradiation or greyscale lithography for the low-cost fabrication of reconfigurable polymeric micromolds to create large area PDMS micropatterns by standard soft-lithography potentially exploitable for different fields of application mainly including microfluidics and microoptics. The real possibility to introduce novel smart fabrication process capable of decreasing time and costs typically required by standard lithography while maintaining the same resolution has a strong soundness in the micro/nanofabrication fields as also demonstrated by the big research efforts focusing on the smart combination of 3D-printing with novel materials. Undoubtedly, the approach reported is very appealing, it uses low-cost materials and enables for micromolds easily replicable with PDMS.

Answer: We thank the reviewer for the positive assessment of our work. We would merely like to point out, that our approach is not based only based on photomasks but can be used with maskless projection lithography as well (as shown in the manuscript).

Question 8. Fastness: *The main point I would like to highlight is that the authors stated that the reported fabrication process is time-saving, especially considering that reconfigurable molds can be used to create many replicas. According to what reported in the Manuscript, summing up the times reported in the materials and methods section, at least six days are necessary to fabricate the polymer molds, engraving it, replicating with PDMS and reconfiguring the molds. Furthermore, these 6 days could be acceptable if the molds could be replicated many times, while the authors stated that they can be used for 4 replicas before degrading. Also from Figure 2 E, from the 2nd to*

the 4th replicas the shape differences as well as the different surface roughness are evident.

Answer 8: We understand the reviewer's concern. However, the appeal of our method is that the materials can be conveniently reused, thus, once prepared can be used for several replications. To emphasize this, we have included data that shows that the gels can be conveniently stored for months and be used without significant loss in performance.

Additionally, we have executed multiple structuring and erasing cycles showing that there is no noticeable fatigue in the gel.

We have made the following changes to the manuscript

We have included the following sentences into results to show hydrogels' stability after being stored for 5 month:

In addition, the prepared hydrogel can be stored for long time in the glass cell (at least 5 months) without noticeable loss the performance, which is desirable in practical applications. To verify this, a micro-square array was structured onto the freshly prepared hydrogel and the hydrogel stored for 5 months using a physical mask (illumination 10s), the obtained structures show no difference (Supporting information Fig. 8).

We have included the following data into supporting information:

Supplementary Fig.8 WLI images of PDMS microstructures replicated from a fresh (a) and a five-month stored (b) hydrogel, and (c) corresponding statistical analysis of height distribution.

We have included the following sentences into the results to show hydrogels' stability after multiple structuring-erasing cycles without PDMS replication:

In addition, as shown in supplementary Fig.11, the hydrogel shows no significant fatigue after 6 times structuring-erasing cycles, which shows its stable ability to be restructured as micromolds.

We have included the following figure to supplementary information:

Supplementary Fig.11 Multiple structuring-erasing cycles on the hydrogel without PDMS replication. (a-c) WLI images of structured hydrogel surface and (g) profiles of corresponding microstructures.

Question 9: *It is also important to take into account that the main proof-of-concepts demonstrating the potential applications of the reconfigurable molds for microfluidics and microlens arrays i) request the use of a physical mask to guarantee the resolution and ii) are obtained by two-steps PDMS replicas, where the 1st PDMS replica from the reconfigurable molds thus requiring an additional step of silanization and further extending fabrication times.*

Answer 9: We understand the reviewer's concern. As an alternative, we have included a novel microfluidic chip that directly generates the negative of the chip and only requires a single replication step. Furthermore, to emphasize the strongpoint of the method, which is the generation of greyscale images, we have included the generation of a complex DOE structure by maskless lithography – emphasizing that the maskless illumination also provided sufficient resolution for optical applications (see Answer2 Fig. 4d, e and f).

We have made the following changes to the manuscript:

We have included the following sentences into results:

As an alternative, a microfluidic chip can be generated only requiring a single replication step which shows same performance although with lower depth (Supplementary Fig. 17).

We have included the following figure into the supplementary information:

Supplementary Fig.16. Fabrication of microchannel using a negative mask. (a) The negative mask employed for engraving convex microstructure on the hydrogel surface; (b) The optical image of hydrogel with convex microstructure; (c) WLI characterization and (d) feature profile of replicated microchannel in

PDMS at different locations along the channel; (e) Optical picture of the microchannel filled with dyed water showing diffusive mixing.

Question 10: *In addition, I have some doubts on the possibility to crosslink the PDMS at room temperature in only 6 hours. Standard procedure typically requires at least 24 hours when working at ambient temperature thus further increasing time for replica preparation.*

Answer 10: The PDMS is not completely cured after 6 hours, but after 6 hours, the PDMS already was rigid enough to lift it off the hydrogel surface and conduct experiments/measurements.

Question 11. *Low cost procedure: If I well understood, to demonstrate the potential application of the reconfigurable molds for high-resolution microfluidic and microoptics, the engraving process is based on the UV-irradiation through physical masks. Are these masks fabricated by standard processes? If this is the case, then the presented approach still relies on side expensive masks fabrication processes when higher resolution is needed.*

Answer 11: We validated the process using physical masks printed by KOENEN GmbH, 155 Euros including tax, package and delivery fee as well as using maskless projection lithography which has no overheads as no physical masks are required.

Question 12. *High-resolution: the authors stated that they can exploit the procedure for high-resolution polymer replication at lower costs. However, for proof-of-concept applications where the highest resolution is obtained, mask-assisted UV irradiation is needed, and I was wondering how these masks are*

produced. On the other hand, the resolution decreases when using DMD and digital masks especially with respect to reported literature and considering fabrication resolution that can be obtained by replica molding of polymeric stamps realized by effective low-cost rapid prototyping methods such as 3D printing. Also I am not so convinced on the surface roughness of the replicas as well on the exact shape of the arrays obtained by using the grayscale projection. I would suggest adding SEM images at least to each example of applications as reported for the microlens arrays.

I would like the authors to comment on these three aspects that I consider critical.

Answer 12: We are happy to elaborate.

(1) As stated in Answer 11 the masks are simple foil-printed masks and not made by lithography techniques but high-resolution toner printing. This makes them cheap and easily accessible. The resolution of the DMD device is limited by the performance of the optics stack and can be as high as 600 nm per pixel as previously reported (Waldbaur et al., CAMF, 2011)

(2) 3D Printing as an alternative for the fabrication of intricate structures is feasible, however polymeric stamps with smooth surface can be obtained only with very high-resolution technologies like two photon 3D polymerization which requires very expensive instruments and only affords relatively small lateral sizes.

(3) We have included SEM images of greyscale image shown in Fig. 3e, 3f, 3h, and 3k. to supporting information.

We have made the following changes to the manuscript:

We have included the following sentence into the results:

The SEM images of obtained microstructures show decent surface quality (Supplementary Fig. 13).

We have included the following figure to supplementary information:

Supporting Fig. 13 SEM images of samples prepared via maskless grayscale projection lithography shown in Fig. 3e, 3f, 3h, and 3k.

Question 13. *What is the difference in wettability between the cis and trans form? I think should be something already known in literature.*

Answer 13: Difference in wetting ability between the cis and trans form of azobenzene has been reported. Due to the decrease of dipole moment from about 3.1 D (cis form) to 0.5 D (trans form), the azobenzene at cis form is more

hydrophobic than the azobenzene at trans form (see reference 41, 42). Utilizing this property, a smooth monolayer containing azobenzene on silicon substrates shows different water contact angle, decreasing from $69 \pm 2^\circ\text{C}$ (*cis* form) to $62 \pm 2^\circ\text{C}$ (*trans* form) (see reference 43).

We have made the following changes to the manuscript:

We have included the following change to the introduction:

As an example, Kuenstler *et al.*⁴⁰ fabricated a photoresponsive hydrogel based on the host-guest complex between cyclodextrins and azobenzene groups which shows reversible wetting property changes owing to isomerization⁴¹⁻⁴³.

We have included the following papers to the reference:

- 41 Sarkar, N., Bhattacharjee, S. & Sivaram, S. Surface Functionalization of Poly(ethylene) with Succinic Anhydride: Preparation, Modification, and Characterization. *Langmuir* **13**, 4142-4149, doi:10.1021/la9610664 (1997).
- 42 Raduge, C., Papastavrou, G., Kurth, D. G. & Motschmann, H. Controlling wettability by light: illuminating the molecular mechanism. *Eur Phys J E Soft Matter* **10**, 103-114, doi:10.1140/epje/e2003-00015-0 (2003).
- 43 Delorme, N., Bardeau, J. F., Bulou, A. & Poncin-Epaillard, F. Azobenzene-Containing Monolayer with Photoswitchable Wettability. *Langmuir* **21**, 12278-12282, doi:10.1021/la051517x (2005).

Question 14. *Based on experience of PDMS replicas of plastic molds, I am aware that when replicating from hydrogels or materials not completely cured, there could be a sticky effect that affects PDMS polymerization or that could require longer PDMS curing (again in contrast with the 6h curing process mentioned in 1)). Probably the authors observed this phenomenon as stated in lines 167-168. Furthermore, I would like to ask if the percentage of crosslinker could affect the speed of reconfiguration.*

Answer 14:

(1) The reviewer is right, this is a common problem, e.g. in 3D printed molds, especially on hydrophobic materials. But for the herein reported hydrogel, this is not the case due to the high hydrophilicity and thus, the limited adsorption to the hydrophobic PDMS. It has been previously reported that PDMS can cure well on the water or hydrogels (Kim, D.et al. (2019). doi:10.3390/polym11081264; Dang, T.et al. (2012). doi:10.1088/0960-1317/22/1/015017).

(2) For the phenomenon as stated in lines 167-168, we revised the text to make this clearer:

~~However, higher softness prevents the effective separation of the cover glass slide (see Supplementary Fig.5), which cause damage to the gel surface. In contrast, higher crosslinking (up to 2 mol% of crosslinker) allows gels with higher elastic modulus and improved mechanical resilience, but reduced engraving speed (Fig. 2b). As a compromise, a gel with 1.6 mol% crosslinker was chosen.~~

However, higher softness causes the hydrogel vulnerable to adhesion force between the hydrogel and cover slide when opening the glass cell (Supplementary Fig.5) in which the gel was polymerized, which relatively makes it more difficult to obtain hydrogel with high surface quality. In contrast, higher crosslinking density (up to 2 mol% of crosslinker ratio) allows gels with higher elastic modulus and improved mechanical resilience, but reduced engraving speed (Fig. 2b). As a compromise, a gel with 1.6 mol% crosslinker ratio was chosen.

Supplementary Fig. 5 Setup for the AM/AZO-CD gel polymerization.

(3) The reviewer is right, adjusting the crosslinker ratio influences the response speed of the hydrogel. We have shown different crosslinker ratios in Figure 2a. As we have shown in Fig. 2b, the softer gels with less crosslinker made deeper structures in short time frames. Thus, the writing speed and erasing speed is influenced by the crosslinker. The very soft gels were not suitable for replication, which is why a “slower” gel was chosen for the application experiments.

Question 15. *Could you comment on the trend obtained in Figure 3c?*

Answer 15: We assume that the reviewers is referring to the curve in Fig. 3c showing a relatively high growth rate for the first two minutes, but after that, the growth rate slows down.

We have included the following changes to the draft:

We have included the following sentences to the results:

Owing to the limited UV penetration ability, the very top hydrogel surface receives the highest exposure energy to induce the isomerization of azobenzene groups and results in rapid growth speed of the engraved structure. With increasing depth into the hydrogel, the received exposure energy reduces gradually, and the growth speed decreased correspondingly. In addition, the contraction of the exposed area is constrained by the ambient polymer network, the constrain force increases with larger contraction, which also contribute the

slower growing speed with increasing exposure time. Similar phenomenon has been reported when fabricating microstructures on a soft photo responsive substrate^{26,27}

We have included the following papers to the reference:

26 Chen, D. *et al.* Homeostatic growth of dynamic covalent polymer network toward ultrafast direct soft lithography. *7*, eabi7360, doi:doi:10.1126/sciadv.abi7360 (2021).

27 Li, T. *et al.* Hierarchical 3D Patterns with Dynamic Wrinkles Produced by a Photocontrolled Diels–Alder Reaction on the Surface. *32*, 1906712, doi:10.1002/adma.201906712 (2020).

Question 16. *The caption 4a and 4b should be inverted, and at line 223 the reference for figure 4f should be changed with 4c.*

Answer 16: The mistakes have been corrected.

The captions of 4a and 4b have been corrected (see Answer2)

We have included the following change to correct the reference of figure 4c

The replicated PDMS microlens array shows uniform structures (Fig. 4a & b) and the intended clear diffraction pattern (laser source: 532 nm) (Fig. 4fc)

REVIEWERS' COMMENTS

Reviewer #1 (Remarks to the Author):

The authors have addressed the experimental and analysis concerns raised in my initial review. I am still not convinced that their approach will find broad application for replica molding of PDMS, but they offer an alternative method to an established field of research and it may find use for optical components as proposed by the authors in the revised manuscript.

As a follow-up from the first review, the authors should still update the captions of Figures 2 and 3 to indicate the meaning of the error bars (SD, SEM, CI, ...) in Fig. 2a-c and 3c as well as the number of samples investigated to calculate the error bar for each data point.

As a minor comment, it is still not entirely clear to me how the authors distinguish the need for a clean room to perform light-based patterning of SU-8 or their hydrogel material, since particulate contamination seems to be an equal problem or non-problem in both cases, depending on the required feature sizes and acceptable areal defect rates. It is not a strict requirement, but it would be nice if the authors could comment on why their approach would be more tolerant to ambient particulate contamination than other microfabrication processes for PDMS molding.

Reviewer #3 (Remarks to the Author):

Even if not still convinced on the fastness of the overall procedures, I am satisfied with the response and the corrections. As a minor the caption in Figure 11 in supporting information is not correct (should be a-d instead of a-f). Then, I recommend its acceptance at the current stage.

Dear reviewers, dear editors,

We thank the editors and reviewers for their expert input and detailed review of our work and the helpful suggestions they have made. We have addressed all the issues raised – please find our response to the comments of the reviewers and editors on the following pages. The reviewer's comments are set in italic whereas our comments are set in upright font. Changes to the manuscript are marked in yellow.

Reviewer 1

The authors have addressed the experimental and analysis concerns raised in my initial review. I am still not convinced that their approach will find broad application for replica molding of PDMS, but they offer an alternative method to an established field of research and it may find use for optical components as proposed by the authors in the revised manuscript.

Answer: We thank the reviewer for this positive assessment of our work.

Question1. *As a follow-up from the first review, the authors should still update the captions of Figures 2 and 3 to indicate the meaning of the error bars (SD, SEM, CI, ...) in Fig. 2a-c and 3c as well as the number of samples investigated to calculate the error bar for each data point.*

Answer1: The number of samples investigated to calculate the error bar for each data point has been added to the associated legends (see **Answer6**).

Question2. *As a minor comment, it is still not entirely clear to me how the authors distinguish the need for a clean room to perform light-based patterning of SU-8 or their hydrogel material, since particulate contamination seems to be an equal problem or non-problem in both cases, depending on the required*

feature sizes and acceptable areal defect rates. It is not a strict requirement, but it would be nice if the authors could comment on why their approach would be more tolerant to ambient particulate contamination than other microfabrication processes for PDMS molding.

Answer2: A dust-free environment is always necessary for good results but cannot be compared to the need for a high-end cleanroom with large machinery. Our method uses simple chemicals, simple casting and illumination techniques and can be done in many labs. A dust-free environment can also be achieved by a simple box.

Reviewer 3

Even if not still convinced on the fastness of the overall procedures, I am satisfied with the response and the corrections.

Answer: We thank the reviewer for this positive assessment of our work.

Question 3. *As a minor the caption in Figure 11 in supporting information is not correct (should be a-d instead of a-f). Then, I recommend its acceptance at the current stage.*

Answer3: We thank the reviewer for spotting this - the mistake has been corrected.

We have included the following change to supporting information.

Supplementary Fig.11 Multiple structuring-erasing cycles on the hydrogel without PDMS replication. (a-f) WLI images of structured hydrogel surface and (g) corresponding profiles of corresponding microstructures.

In line with the requirements listed in author checklist, we have included the following changes to the draft.

Question4. *The final paragraph of the Introduction must begin with a phrase like “In this work” or “Here, we show”, and contain a brief summary of the major results and conclusions of the current work, written in the present tense.*

Answer4: we have included the following changes to the introduction.

In this **paper work**, we present a facile method to generate micromold displays based on the photoresponsive acrylamide/azobenzene-cyclodextrin (AM/AZO-CD) hydrogel prepared via thermal radical polymerization and their usage in polymer replication.

Question5. *Please divide the Results section into subsections, each with a title of 60 characters or fewer including spaces.*

Answer5: we have included the following changes to the result part.

~~Micromold display structures using high-resolution maskless grayscale projection lithography~~

Micromold display via grayscale projection lithography

Question6. *The sample size (n) must be stated in the corresponding figure legend and data points should be shown for plots with $n < 10$. For larger sample sizes, please consider box-and-whisker or violin plots as alternatives. Measures of centrality, dispersion and/or error bars should be plotted and described in the figure legend.*

Answer6: we have included the following changes to the Fig. 2 and Fig. 3.

Fig. 2 Optimization and usage of the micromold display. (a) Increasing crosslinker ratio results in AM/AZO-CD hydrogels with a higher elastic module; (b) Higher crosslinker ratio causes a decrease of the engraving speed on the hydrogel, but facilitate handing of the micromold displays; consequently, the hydrogel formulation with 1.6 mol% crosslinker was chosen for further experiments; (c) Engraving height increases with increasing exposure time, reaching 10.6 μm after 15 min UV irradiation; (d) Illustration of consecutive setting/resetting cycles of the micromold display with different topographies; (e) Optical pictures and WLI characterization of four PDMS substrates replicated from micromolds generated on the micromold display with a

smiley (1st and 3rd) and a deer (2nd and 4th) structure. Data in a, b and c are presented as mean values \pm SD. Error bars represent the standard deviation from three samples.

Fig. 3 Micromold display structures generated using digital light processing (DLP). (a) Scheme of maskless projection lithography system based on a digital micromirror device (DMD); (b) WLI profile of replicated PDMS with a triangle array; (c) Engraving height increases over exposure time reaching 11 μm after 10 min of

illumination; (d) The profile of replicated PDMS with a tree structure showing the ability of the micromold display to fabricate complex shapes; (e & f) Two consecutive lithography-replication process based on reversibility of the AM/AZO-CD hydrogel via digital masks using the custom-made DMD; (g) A square array greyscale digital mask used for grayscale lithography, and (h & i) WLI image and feature profile of obtained structure on the replicated PDMS with different height; (j) A grayscale flower mask used for grayscale lithography and (k & l) WLI image and feature profile of replicated flower on PDMS with different height based on the varied transparency of the mask. Data in c are presented as mean values \pm SD. Error bars represent the standard deviation from three samples.

Question7. *Should you choose to share your datasets or source data using our Figshare integration. Please ensure the manuscript references the Figshare DOI link in its Data Availability Statement. In this case, please include the Figshare to your reference list following the format "authors, title, Figshare, DOI, year" and include the following statement to your Data Availability Statement to:*

"Source Data file has been deposited in Figshare under accession code DOI link.[#]"

Answer7: We have included the following change to the data availability statement.

Data availability. ~~All data reported in this paper are available from the authors on request.~~ Source Data file has been deposited in Figshare under accession code DOI link: <https://doi.org/10.6084/m9.figshare.25726524>.